# A Simple Nomogram to Predict Clinically Significant Prostate Cancer at MRI-Guided Biopsy in Patients with Mild PSA Elevation and Normal DRE

**DOI:** 10.3390/cancers17050753

**Published:** 2025-02-23

**Authors:** Hubert Kamecki, Andrzej Tokarczyk, Małgorzata Dębowska, Urszula Białończyk, Wojciech Malewski, Przemysław Szostek, Omar Tayara, Stefan Gonczar, Sławomir Poletajew, Łukasz Nyk, Piotr Kryst, Stanisław Szempliński

**Affiliations:** 1Second Department of Urology, Centre of Postgraduate Medical Education, 01-809 Warsaw, Poland; 2Nałęcz Institute of Biocybernetics and Biomedical Engineering, Polish Academy of Sciences, 02-109 Warsaw, Poland

**Keywords:** prostate cancer, csPC, nomogram, fusion biopsy

## Abstract

In this retrospective study of 209 patients, we developed a simple model to calculate the probability of diagnosing clinically significant prostate cancer (csPC) at MRI–ultrasound fusion biopsy in men with low suspicion of the disease (prostate specific antigen < 10 ng/mL; normal digital rectal examination) but positive magnetic resonance imaging findings. The model demonstrated good discrimination performance, consistent on internal validation. Using an empirical threshold of <10% csPC probability to omit biopsy, half of unnecessary biopsies could have been avoided, with only two (3.0%) csPC cases missed. Pending external validation, this model could be a valuable tool for reducing unnecessary biopsies.

## 1. Introduction

In recent decades, prostate-specific antigen (PSA) testing has emerged as a widely used screening tool for prostate cancer (PC), yet its extensive use has inadvertently led to notable rates of PC overdiagnosis [1]. To mitigate this effect by primarily identifying clinically significant PC (csPC) cases, multiple tools have been introduced, including magnetic resonance imaging (MRI) of the prostate and the Prostate Imaging Reporting and Data System (PIRADS) classification.

The current European Association of Urology (EAU) guidelines recommend that patients with mild PSA elevation (3–10 ng/mL) and a normal digital rectal examination (DRE) should be assessed with either a valid risk calculator or an MRI of the prostate, with PIRADS ≥ 3 lesions indicating a need for a biopsy [2]. Nevertheless, a notable proportion of patients with positive MRI findings will not be diagnosed with csPC [3]. Thus, obtaining evidence helpful in avoiding unnecessary biopsies and low-risk PC overdiagnosis appears to be essential.

The aim of this study was to examine the institutional cohort of patients with relatively low suspicion of csPC (i.e., PSA < 10 ng/mL and a normal DRE) who underwent MRI-guided software fusion biopsy of the prostate due to positive MRI findings (PIRADS ≥ 3). Our primary objective was to develop and internally validate a nomogram for predicting csPC in such a setting.

## 2. Materials and Methods

We retrospectively analyzed consecutive patients who underwent MRI–ultrasound fusion biopsy of the prostate at the Second Department of Urology of the Centre of Postgraduate Medical Education between November 2019 and October 2022. Data were collected from medical patient records and included age, past medical history, pre-biopsy PSA level, MRI report, biopsy procedure report, and pathology report. Inclusion criteria were as follows: (i) pre-biopsy PSA level < 10 ng/mL, (ii) normal DRE at biopsy, and (iii) PIRADS category ≥ 3. Patients were excluded if any of the following were present: (i) prior history of PC, (ii) antiandrogen or androgen deprivation therapy, (iii) extraprostatic extension (EPE) or seminal vesicle invasion (SVI) at MRI, or (iv) missing or incomplete data.

### 2.1. MRI–Ultrasound Fusion Biopsy

The patients underwent MRI either at our institution or externally. Prior to biopsy, external studies might have been reviewed by an institutional radiologist in case of ambiguities. All lesions were assessed using PIRADS version 2.0 or 2.1. Biopsies were performed using either transrectal or transperineal approach. MRI–ultrasound software fusion was applied in all cases. All procedures were conducted by a single experienced urologist. While every biopsy included targeted cores, systematic cores might have been omitted in selected cases. Additional cores targeted at lesions considered suspicious by the performing urologist or the reviewing radiologist might have been taken. Systematic cores were intended not to cover the regions subject to targeted biopsy. All specimens were assessed by institutional pathologists specializing in prostate cancer.

A sample picture documenting a typical MRI–ultrasound fusion biopsy, performed in a patient included into this study, is presented in Appendix A.

### 2.2. Definitions

We defined highest PIRADS category as the highest category assigned in the original MRI report, regardless of the reviewing institutional radiologist’s second opinion. This approach was aimed at representing a typical clinical scenario. Maximum lesion size was the maximal diameter of the largest lesion of the highest PIRADS category in the original MRI report. Prostate volume (PV) was the volume provided in the MRI report. Lesion involving the peripheral zone (PZ) was any PIRADS ≥ 3 lesion involving PZ in the patient. We defined csPC as grade group ≥ 2 cancer. Number of lesions was calculated for PIRADS ≥ 3 lesions only.

### 2.3. Outcome Measurements and Statistical Analysis

Categorical and quantitative variables were expressed as numbers with percentages and medians with interquartile ranges (IQRs), respectively. Associations between potential risk factors and the dependent variable (csPC) were first investigated using univariable logistic regression analysis. Factors exhibiting statistically significant associations on univariable analysis were then considered for multivariable logistic regression analysis. Simultaneously, the stepwise method was applied to identify potential significant factors for a multivariable model. To avoid interdependencies within a model, separate models, which included either PSA and PV, or PSA density (PSAD), were developed, and the model with higher discrimination ability, as demonstrated with receiver-operating characteristic (ROC) area under the curve (AUC), was selected for further consideration. Following the multivariable logistic regression analysis, a nomogram was constructed and the predicted probability of csPC was calculated using the equation computed with the regression model.

We performed an internal validation of the model using 5-fold cross-validation, repeated 500 times, and bootstrapping, with 100 resampling iterations. We developed calibration curves. The discrimination ability, diagnostic performance, and clinical utility of the model fitted on the whole training set and the models fitted during validation process were calculated. For diagnostic performance analysis, the default threshold (0.50) was used.

Associations were considered statistically significant if *p*-value < 0.05. All statistical analyses were performed using R version 4.2.0 (R Foundation for Statistical Computing, Vienna, Austria). Nomogram was prepared using the function ‘nomogram’ from the ‘rms’ package. Construction and validation of the model was performed using the ‘caret’ package.

## 3. Results

We identified 262 patients who met the inclusion criteria. We excluded 18 and 35 patients due to prior history of PC and missing or incomplete data, respectively (see the flowchart in Figure 1); none of the patients met other exclusion criteria. Eventually, 209 patients were included in the analyses. Patient characteristics are presented in Table 1.

On univariable analysis, age, PSA, 5-alpha reductase (5-ARI) use, PV, PSAD, PIRADS > 3, maximum lesion size, and lesion involving PZ demonstrated statistically significant associations with csPC (Table 2). Models incorporating those variables were developed; however, maximum lesion size did not demonstrate a statistically significant association with csPC upon multivariable analysis and thus was removed from the models. Stepwise analysis led to the development of identical models. The results of the multivariable analysis are presented in Table 2.

The model incorporating PSA and PV demonstrated higher ROC AUC and thus was selected for further analyses. Based on the coefficients computed with the multivariable regression model, the predictive model was developed, as follows:(1)$$csPC=\frac{1}{1+e^{-MVA}}$$(2)$$MVA=-7.8332+0.06\times \left[age\right]+1.4825\times \left[5ARI\;use\right]+0.4068\times \left[PSA\right]-0.0517\times \left[PV\right]\;+1.8452\times \left[PIRADS>3\right]+1.5614\times [PZ]$$
where [*age*] is the patient’s age (years attained), [*5ARI use*] is the use of 5-ARI (yes = 1, no = 0), [*PSA*] is the pre-biopsy serum PSA level (ng/mL), [*PV*] is PV (mL), [*PIRADS* > 3] is the maximum PIRADS category > 3 (yes = 1, no = 0), and [*PZ*] is any PIRADS ≥ 3 lesion involving the peripheral zone (yes = 1, no = 0). The nomogram, based on the predictive model, is shown in Figure 2.

The predictive model showed good discrimination ability (ROC AUC = 0.86 with the model fitted on the whole training set) and the performance metrics were consisted after internal validation of the model with 5-fold cross-validation and bootstrap methods (Table 3, Figure 3).

Calibration curves from both the full training set and the validation process demonstrated good calibration (Figure 4). Decision curve analysis demonstrated a pronounced net benefit for the nomogram, as compared with curves for individual risk factors (Figure 5).

Multiple different cut-off values of the predictive model were evaluated for clinical utility on the whole training set (Table 4). With an empirical threshold of 10% csPC probability, omitting biopsy would result in avoiding 72 (50.7%) unnecessary biopsies at the cost of missing 2 (3.0%) csPC cases.

## 4. Discussion

We present a novel nomogram designed to predict csPC at MRI-guided biopsy. The model demonstrated good discrimination, confirmed through internal validation with bootstrap and 5-fold cross-validation.

Numerous csPC risk calculators have been already published in the literature, including the widely recognized ERSPC calculator [4]), or the most recent works by Wagaskar et al. [5], Zhou et al. [6], Liu et al. [7], and Parekh et al. [8]. However, our study offers a model developed specially for a population of patients deemed to be at low risk of harboring csPC, i.e., men with PSA < 10 ng/mL, negative DRE, and an organ-confined lesion at MRI. Those patients, often considered “equivocal” in clinical practice, may particularly benefit from reduced unnecessary biopsy rates. Another significant strength of our nomogram is its simplicity and use of easily accessible clinical parameters, making it practically applicable for nearly all patients encountering biopsy decisions. Moreover, our study exclusively involved patients undergoing MRI-guided biopsy of the prostate, now considered the gold standard and strongly advocated by the guidelines [2]. All biopsies were performed by an experienced operator, which further underscores the reliability of our findings [9].

Most variables included in the nomogram are well recognized for their association with an increased risk of detecting csPC. PSA is a traditional serum marker used for PC early detection. Interestingly, according to Palsdottir et al., only csPC may cause PSA elevation exceeding the contribution of non-cancer prostate tissue [10]. The significance of age as a risk factor, especially for higher-grade prostate cancer, is well documented in the literature [11]. Given that higher PIRADS categories are associated with a greater risk of csPC [3], it is unsurprising that PIRADS category > 3 correlates with csPC in our study. We opted not to distinguish between PIRADS 4 and 5 in our model, as lesion size, which is the exclusive parameter differentiating between those two categories in our population (patients with EPE or SVI were not included), did not show statistically significant association with csPC on MVA. Prostate tumors most commonly originate in the PZ, where they tend to be more aggressive than those in the transitional zone [12,13]. This aligns with the significant association observed in this study. Our findings also support the literature indicating a higher risk of csPC detection in smaller prostates [4], likely due to larger prostates being associated with non-cancer-related PSA elevations [14]. We also evaluated PSAD, which adjusts PSA levels based on PV and is an established predictor of csPC at biopsy [15]. To avoid potential confounding in MVA, we chose to consider either PSAD or separate measurements of PSA and PV. The model incorporating PSA and PV demonstrated a higher ROC AUC in our training set and was therefore selected for further analyses.

The association between 5-ARI use and increased risk of csPC is notable. While a possible impact of 5-ARIs on the natural history of prostate cancer remains a subject of debate [16,17], a likely explanation of our results is the well-documented decrease in PSA levels caused by 5-ARI therapy [18]. Many participants who met the study’s inclusion criterion of PSA level < 10 ng/mL might have been classified as higher-risk for csPC if not for their use of 5-ARI. Our results suggest that in 5-ARI-users who present with elevated PSA levels, a lower threshold for clinical suspicion should be applied. 

In our study, maximum lesion size was not found to be associated with the detection of csPC, possibly due to predominance of small tumors in the analyzed group, as detailed in Table 1. To our knowledge, there is no compelling evidence linking the multifocality of PIRADS > 3 lesions to an altered risk of csPC risk, and our findings suggest that such multifocality may not contribute to higher (or lower) risk levels. Interestingly, although previous studies have indicated a lower risk of csPC in patients with prior negative biopsy [3,19], in our group, the link between being biopsy-naïve and csPC was not statistically significant. This absence of significance might be due either to the study being underpowered or to a genuine lack of association in this particular population.

Unfortunately, we were not able to include several parameters helpful in assessing individual pre-biopsy risk of csPC, including PSA-related history such as previous PSA levels [20], PSA velocity [21], or free PSA percentage [22], as well as urine- or blood-based biomarkers [23]. Having incorporated these factors might have further enhanced the performance of our model.

While we aimed for our nomogram to include only easily accessible features, thereby facilitating clinical utility without the need for additional diagnostic procedures or specialized software, the rapidly emerging role of radiomics in pre-biopsy risk stratification cannot be overlooked. Current evidence suggests that incorporating radiomic features into our model might have further improved its diagnostic performance [24,25].

The main limitation of this study is its retrospective character. For this reason, a risk of significant selection bias arises, as there were no predefined criteria for a patient having been offered a biopsy. Patients who had undergone biopsy, hence included into the study, were men selected for biopsy based on the urologist’s suspicion of csPC, which may have not been solely based on the variables assessed in this manuscript. Hypothetically, there might have been two patients, identical in terms of the variables assessed in the study, but different in regard to other, non-assessed factors. Due to those non-assessed factors, the one with higher suspicion of csPC was offered a biopsy and diagnosed with csPC, and the other one, hypothetically not harboring csPC, did not undergo a biopsy. It is hard to predict the direction of this bias, but one may assume that our model may tend to overestimate the risk of csPC if extrapolated on the general population.

The use of non-institutional MRI reports and the lack of standardized procedures for ensuring consistency in PIRADS scoring may be considered a limitation of this study, given the known interobserver variability in this area [26]. On the other hand, our approach may better represent a real-world outpatient scenario. Pathology specimens were not reviewed for the purpose of this study, and the reassignment of cancer grade at secondary pathology review is a well-described issue [27]. Also, no specific biopsy template was used in our patients; however, the impact of a particular template in patients undergoing MRI–ultrasound fusion biopsy of the prostate may be negligible [28].

## 5. Conclusions

We present an internally validated nomogram that demonstrates good performance in predicting csPC diagnosis at MRI-fusion prostate biopsy in men deemed to be at low risk of harboring the disease. However, the limitations of the study, mainly resulting from its retrospective character, warrant cautious interpretation of the results. External validation of our model is required before the nomogram can be applied in clinical practice.

## Figures and Tables

**Figure 1 cancers-17-00753-f001:**
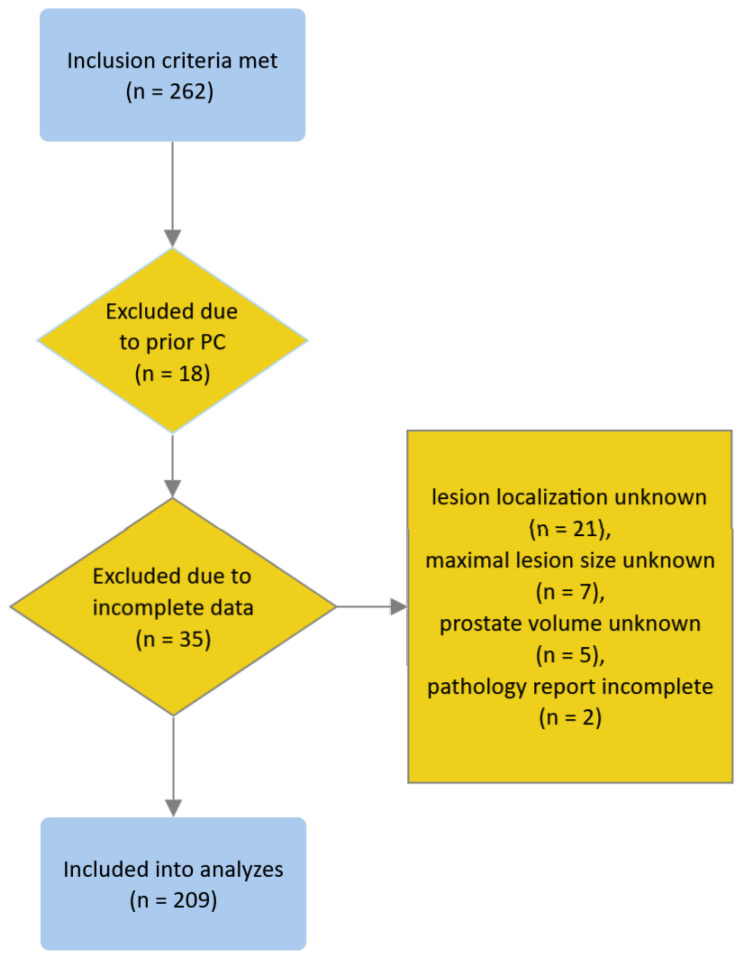
Flowchart depicting the process of patient inclusion.

**Figure 2 cancers-17-00753-f002:**
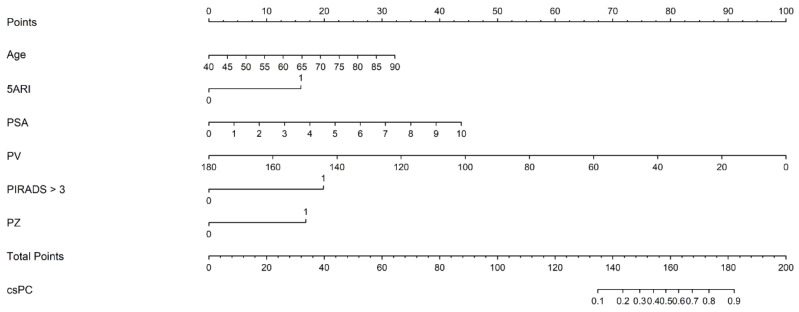
The nomogram based on the predictive model of clinically significant prostate cancer probability.

**Figure 3 cancers-17-00753-f003:**
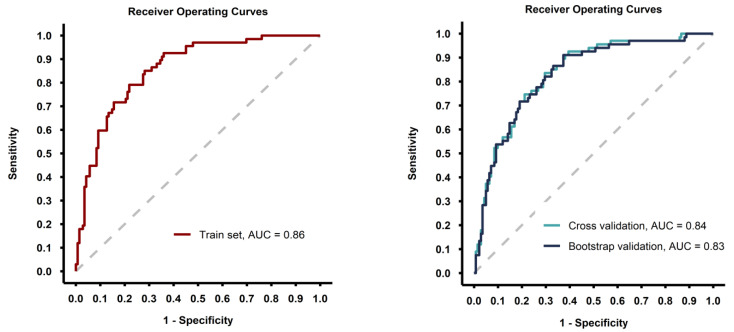
Receiver-operating curves for the model fitted on the whole training set (**left image**) and for the models fitted during the validation process (**right image**).

**Figure 4 cancers-17-00753-f004:**
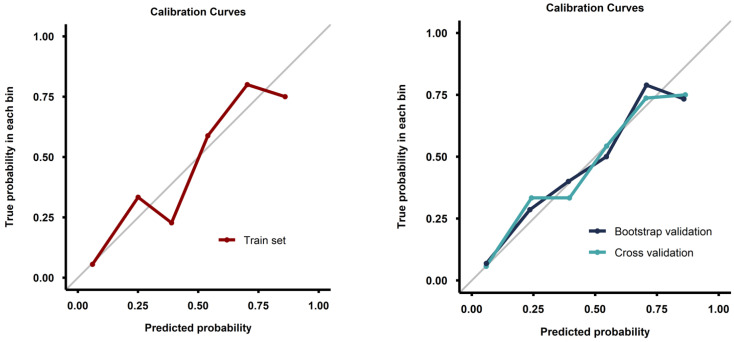
Calibration curves for the model fitted on the whole training set (**left image**) and for the models fitted during the validation process (**right image**).

**Figure 5 cancers-17-00753-f005:**
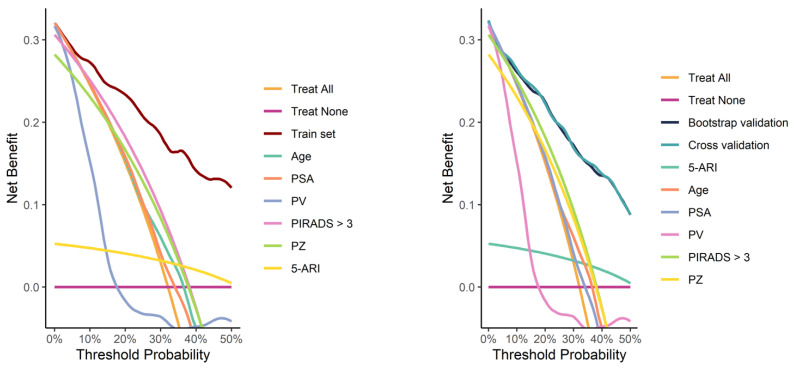
Decision curves for the model fitted on the whole training set (**left image**) and for the models fitted during the validation process (**right image**), compared with curves for individual risk factors.

**Table 1 cancers-17-00753-t001:** Patient characteristics.

Characteristic		All Patients (n = 209)
Median age, years (IQR)		65 (58–70) *
Biopsy-naïve (%)		169 (80.9%)
5-ARI use (%)		21 (10.0%)
PSA, ng/mL (IQR)		5.7 (4.6–7.7) **
PV, mL (IQR)		42 (33–54) **
PSAD, ng/mL^2^ (IQR)		0.13 (0.09–0.18) **
Number of lesions (%)	1	109 (52.2%)
	2	72 (34.4%)
	3	28 (13.4%)
Maximum PIRADS (%)	3	41 (19.6%)
	4	127 (60.8%)
	5	41 (19.6%)
Maximum lesion size, mm (IQR)		12 (8–15) **
Lesion involving PZ (%)		156 (74.6%)
Approach (%)	TR	11 (5.3%)
	TP	198 (94.7%)
Number of cores (IQR)	total	10 (9–12)
	targeted	4 (4–7)
	systematic	5 (4–6)
Any PC (%)		121 (57.9%)
Grade group (%)	1	54 (25.8%)
	2	53 (25.4%)
	3	7 (3.3%)
	4	5 (2.4%)
	5	2 (1.0%)
csPC (%)		67 (32.1%)

IQR, interquartile range; 5-ARI, 5-alpha reductase inhibitor; PSA, prostate specific antigen; PV, prostate volume; PSAD, PSA density; PIRADS, Prostate Imaging Reporting and Data System; PZ, peripheral zone; TR, transrectal; TP, transperineal; PC, prostate cancer; csPC, clinically significant. * normal distribution, *p* = 0.105 at Shapiro–Wilk test; ** not-normal distribution, *p* < 0.001 at Shapiro–Wilk test.

**Table 2 cancers-17-00753-t002:** Univariable and multivariable analyses demonstrating associations between potential risk factors and csPC.

Variable	UVA vs. csPC OR (95% CI), *p*-Value	MVA vs. csPC OR (95% CI), *p*-Value	MVA vs. csPC OR (95% CI), *p*-Value
		AUC = 0.86	AUC = 0.85
Age, years	1.07 (1.03–1.12), <0.001	1.06 (1.01–1.11), 0.014	1.07 (1.02–1.12), 0.006
Biopsy-naïve	0.74 (0.36–1.53), 0.413	– ^a^	– ^a^
5-ARI use	2.59 (1.04–6.49), 0.041	4.40 (1.23–15.74), 0.022	3.87 (1.12–13.4), 0.032
PSA, ng/mL	1.32 (1.13–1.54), <0.001	1.50 (1.23–1.83), <0.001	N.A.
PV, mL	0.97 (0.95–0.99), <0.001	0.95 (0.93–0.97), <0.001	N.A.
PSAD, ng/mL^2^ × 100	1.15 (1.09–1.21), <0.001	N.A.	1.15 (1.09–1.22), <0.001
Number of lesions	0.91 (0.60–1.39), 0.672	– ^a^	– ^a^
PIRADS > 3	7.79 (2.29–26.48), <0.001	6.33 (1.6–25.02), 0.008	5.32 (1.38–20.48), 0.015
Max. lesion size, mm	1.06 (1.01–1.11), 0.030	– ^a^	– ^a^
Lesion involving PZ	3.42 (1.50–7.80), 0.003	4.77 (1.72–13.17), 0.002	4.54 (1.67–12.35), 0.003

UVA, univariable analysis; csPC, clinically significant prostate cancer; OR, odds ratio; CI, confidence interval; AUC, area under the curve; 5-ARI, 5-alpha reductase inhibitor; PSA, prostate-specific antigen; PV, prostate volume; PSAD, PSA density; PIRADS, Prostate Imaging Reporting and Data System; PZ, peripheral zone; N.A., not applicable. ^a^ Variable not included into the MVA.

**Table 3 cancers-17-00753-t003:** Discrimination ability and diagnostic performance of the model fitted on the whole training set and for the models fitted during validation process.

Dataset	ROC AUC	Sensitivity	Specificity	PPV	NPV	Accuracy
Train set	0.86	0.87	0.63	0.83	0.70	0.79
Cross-validation	0.84 ± 0.06	0.85 ± 0.07	0.61 ± 0.13	0.82 ± 0.05	0.67 ± 0.11	0.77 ± 0.06
Bootstrap	0.83 ± 0.05	0.86 ± 0.06	0.59 ± 0.12	0.81 ± 0.05	0.67 ± 0.10	0.77 ± 0.05

ROC AUC, receiver-operating characteristic area under the curve; PPV, positive predictive value; NPV, negative predictive value.

**Table 4 cancers-17-00753-t004:** Analysis of the nomogram-derived thresholds used to differentiate between men with or without csPC. Percentages are calculated in relation to the total number of unnecessary biopsies and the total number of csPC cases.

Threshold (csPC Probability According to the Nomogram)	Below Threshold (Biopsy Not Recommended)	Below Threshold Without csPC (Unnecessary Biopsy Omitted)	Below Threshold with csPC (csPC Missed)	Above Threshold (Biopsy Recommended)	Above Threshold Without csPC (Unnecessary Biopsy Performed)	Above Threshold with csPC (csPC Not Missed)
1%	12	12 (8.5%)	0	197	130 (91.5%)	67 (100%)
2%	16	16 (11.3%)	0	193	126 (88.7%)	67 (100%)
3%	21	21 (14.8%)	0	188	121 (85.2%)	67 (100%)
4%	30	30 (21.1%)	0	179	112 (78.9%)	67 (100%)
5%	42	41 (28.9%)	1 (1.5%)	167	101 (71.1%)	66 (98.5%)
6%	51	49 (34.5%)	2 (3.0%)	158	93 (65.5%)	65 (97.0%)
7%	58	56 (39.4%)	2 (3.0%)	151	86 (60.6%)	65 (97.0%)
8%	62	60 (42.3%)	2 (3.0%)	147	82 (57.7%)	65 (97.0%)
9%	68	66 (46.5%)	2 (3.0%)	141	76 (53.5%)	65 (97.0%)
10%	74	72 (50.7%)	2 (3.0%)	135	70 (49.3%)	65 (97.0%)
11%	76	74 (52.1%)	2 (3.0%)	133	68 (47.9%)	65 (97.0%)
12%	81	78 (54.9%)	3 (4.5%)	128	64 (45.1%)	64 (95.5%)
13%	83	78 (54.9%)	5 (7.5%)	126	64 (45.1%)	62 (92.5%)
14%	84	79 (55.6%)	5 (7.5%)	125	63(44.4%)	62 (92.5%)
15%	87	82 (57.7%)	5 (7.5%)	122	60 (42.7%)	62 (92.5%)

csPC, clinically significant prostate cancer.

## Data Availability

The data analyzed in this study are available upon request from the corresponding author.

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
