# Peer review of "A Simple Nomogram to Predict Clinically Significant Prostate Cancer at MRI-Guided Biopsy in Patients with Mild PSA Elevation and Normal DRE"

_cancers, 2025, doi:10.3390/cancers17050753_

Round 1

Reviewer 1 Report

Comments and Suggestions for Authors

This paper demonstrates that by using a prediction tool  half of the patients could potentially avoid a biopsy.

It is well-written and easy to follow, and I understood what the authors tried to achieve.

However, honestly, I did not find any novelty or significant contribution in this study. Numerous published papers have already investigated similar approaches, including those utilizing AI-assisted MRI techniques like radiomics. etc 

Moreover, in cases where PSA is greater than 2, MRI would typically be performed regardless, even if the DRE results are normal. why we need a nomogram ?

Additionally, the proposed model cannot be reliably used in clinical practice due to two main limitations: the retrospective nature of the study and the small patient sample size.

I recommend incorporating radiomics features into the study to explore how AI-assisted MRI could enhance the accuracy of the proposed model.

Author Response

Dear Reviewer,

Thank you very much for your positive evaluation of the quality of our paper. We greatly appreciate the time and effort you have devoted to reviewing our work.

We fully understand your concerns in regard to our focus on pre-biopsy risk assessment being a well-explored area in the literature. However, as we mentioned in the discussion section, there are two distinctive advantages that set our model apart from the existing studies.

Firstly, our model is specifically developed for a population of “low-risk” patients. By narrowing the study group to this particular cohort, we achieve a better fit for the linear coefficients, which enhances the accuracy of the logistic regression. Furthermore, by focusing on the group of “equivocal patients”, the population in whom the decision is especially challenging, we provide more significant insights into the diagnostic performance (i.e. it is not difficult to demonstrate high accuracy when discriminating among low- and high-risk patients, but when assessing the risk in a cohort of similar, low-risk men).

Secondly, our model is designed using only easily accessible clinical features. This means that no additional, specialized testing is required beyond what is typically available in routine clinical practice. As a result, the model can be readily applied to virtually any patient being considered for a biopsy, thereby facilitating broad clinical utility without the need for further diagnostic procedures.

Additionally, to the best of our knowledge, this is the first study to both demonstrate the impact of 5-alpha-reductase inhibitors on cancer detection at MRI-guided biopsy and to integrate this factor into a predictive model. We believe this represents a significant contribution to the field.

You raised an important point regarding the true impact of digital rectal examination (DRE) results on clinical decision-making in the context of MRI availability. While we agree that most patients with normal DRE but elevated PSA levels will indeed undergo MRI, our model serves as a valuable tool for further risk stratification in these cases. In clinical practice, there are numerous scenarios where patients present with positive MRI, but the clinical picture is equivocal (e.g., normal DRE results, likely PSA contribution from enlarged prostate, significant LUTS, or the PSA levels are stable over time). In such situations, the decision to proceed with biopsy is not always straightforward. Our study aims to provide additional guidance to clinicians facing these complex decisions. In our opinion, positive DRE would trigger biopsy in almost every case of positive MRI, so the “uncertain” patients may be identified only in the DRE-negative group.

Thank you for drawing our attention to the most important limitations of the study. We acknowledge that the retrospective nature of the study and the relatively small sample size currently prevent the model from being reliably implemented in widespread clinical practice. We have thoroughly discussed these limitations in the manuscript. Accordingly, we have clearly stated that external validation is necessary, and we are hopeful that, following publication, the model will be validated in larger, possibly multicenter studies.

Thank you for your suggestion to incorporate radiomic features into the study. We agree that this is a promising and rapidly evolving area of research. However, incorporating radiomics would require a thorough restructuring of the study design and methodology. In the revised version of the manuscript, we acknowledged the emerging role of radiomics in prostate cancer diagnostics (lines: 245-249).

Note: To correctly go the lines by numbers, please select the “Simple markup” or “No Markup” in MS Word.

Once again, we sincerely thank you for your valuable feedback and constructive suggestions.

Reviewer 2 Report

Comments and Suggestions for Authors

The authors present an internally validated nomogram that is helpful to avoid unnecessary prostate biopsies. After performing 5-fold cross-validation and bootstrapping methods, their results demonstrate that the used nomogram has good performance in predicting csPC diagnosis at MRI-fusion prostate biopsy in men. Also, the applicable range of limitations of the study is comprehensively concluded, and I think the method has potentials to be applied in clinical practice in the future.

The following issues are needed to consider and illstrate

1) I suggest adding a typical MRI image to illustrate the mutual concern and complementarity between its feature information and csPC diagnosis. Merely presenting statistical results is not convincing enough;

2) When discussing the correlation between the diagnosis of csPC and the characteristics of patients or lesions, further explanation and clarification are needed for the unclear description and grading of lesions;

3) What is the characteristic distribution of 262 patients as a sample for research and testing, and is it representative among the same patients? It is recommended to create a table to illustrate the distribution of these samples.

Author Response

Dear Reviewer,

Thank you for your thorough review of our manuscript. We truly appreciate the time and effort you spent on helping us in refining our work. Below we provide detailed responses to each of your comments.

“I suggest adding a typical MRI image to illustrate the mutual concern and complementarity between its feature information and csPC diagnosis. Merely presenting statistical results is not convincing enough”

Thank you for this suggestion. We have added a figure including both MRI picture and MRI-ultrasound fusion 3D model (depicting cores taken during biopsy), together with a short clinical vignette describing nomogram result and csPC diagnosis status. The figure is added as supplementary and referred to in the Methods section (lines 81-82).

Note: To correctly go the lines by numbers, please select the “Simple markup” or “No Markup” in MS Word.

“When discussing the correlation between the diagnosis of csPC and the characteristics of patients or lesions, further explanation and clarification are needed for the unclear description and grading of lesions”

We would like to kindly ask for clarification to ensure we address your concerns accurately. We have rephrased part of our discussion section to more clearly articulate the justification for not distinguishing between PIRADS 4 and 5 lesions (lines 211-214). We hope this revision at least partially addresses your concerns. Any further guidance you could provide on this point would be highly appreciated.

“What is the characteristic distribution of 262 patients as a sample for research and testing, and is it representative among the same patients? It is recommended to create a table to illustrate the distribution of these samples”

Thank you for drawing our attention to this issue. We have updated Table 1 in the revised version of the paper. For quantitative variables, we have added footnotes indicating the character of distribution, as calculated with Shapiro-Wilk test. As most of the variables are distributed not-normally, we use median and IQRs in Table 1.

Once again, thank you very much for your constructive feedback and insightful comments.

Reviewer 3 Report

Comments and Suggestions for Authors

1. The title should emphasize the specificity of the patient cohort, e.g., patients with mild PSA elevation and normal DRE. The abstract succinctly summarizes the study but lacks key details, such as sample size after exclusions and calibration outcomes.

2. The introduction effectively sets the stage but misses a detailed justification for the study’s focus on patients with PIRADS ≥3. 

3. The retrospective design introduces potential selection bias that is not adequately addressed. Provide a sensitivity analysis or discuss how the inclusion criteria might skew generalizability. Missing details on how MRI data were standardized across institutions.  Clarify procedures for ensuring consistency in PIRADS scoring and biopsy methodologies.

4. The results are well presented, but the study does not analyze subgroups, such as biopsy-naïve vs. previously biopsied patients, which could impact outcomes. Include calibration plots to visually demonstrate model performance.

5. The discussion overstates the clinical applicability of the nomogram without external validation. Certain findings, such as the role of lesion size, are dismissed without robust justification. 

6. Figures lack sufficient annotations to explain key trends for readers unfamiliar with the statistical methods. Table 4 is dense and difficult to interpret without prior knowledge of thresholds.

7. The conclusion is overly optimistic regarding clinical implementation without discussing potential drawbacks, such as costs and accessibility of MRI-guided biopsies. Redundancy with the abstract’s concluding remarks.

Comments on the Quality of English Language

 There are occasional grammatical errors, such as inconsistent subject-verb agreement and misplaced modifiers (e.g., Calibration curves for both the model fitted on the whole training set and models fitted during the validation process demonstrated good calibration" could be simplified for better readability).

Certain sentences are unnecessarily complex, which detracts from the manuscript's clarity (e.g., "The association between 5-ARI use and increased risk of csPC is a notable finding" could be simplified to "The association between 5-ARI use and csPC risk is notable").

The manuscript frequently relies on passive voice, which can make statements less direct and impactful (e.g., Calibration curves were developed" could be rephrased as "We developed calibration curves").

Author Response

Dear Reviewer,

Thank you for your thorough assessment of our manuscript. We sincerely appreciate the time and effort you devoted to this review. Below we give our point-to-point responses to your comments.

Note: To correctly go the lines by numbers, please select the “Simple markup” or “No Markup” in MS Word.

  1. The title should emphasize the specificity of the patient cohort, e.g., patients with mild PSA elevation and normal DRE. The abstract succinctly summarizes the study but lacks key details, such as sample size after exclusions and calibration outcomes.

Response: Thank you for your comment. The specificity of the patient cohort, “patients with mild PSA elevation and normal DRE,” is reflected in the title. Thank you for drawing our attention to the study size provided in the abstract; we corrected this by providing the number of patients after exclusions. While we recognize the importance of calibration outcomes, we chose to exclude them from the abstract to maintain clarity, as detailed figures are presented within the manuscript.

  1. The introduction effectively sets the stage but misses a detailed justification for the study’s focus on patients with PIRADS ≥3. 

Response: Thank you very much for raising this point. While we acknowledge that some patients with negative MRI (i.e., PIRADS 1-2) may still harbor csPC, current EAU Guidelines recommend biopsy for men with PIRADS ≥ 3 lesions. Appropriate reference to the guidelines is provided in the introduction section. In our study, we decided to focus on patients considered candidates for biopsy according to these Guidelines.

  1. The retrospective design introduces potential selection bias that is not adequately addressed. Provide a sensitivity analysis or discuss how the inclusion criteria might skew generalizability. Missing details on how MRI data were standardized across institutions.  Clarify procedures for ensuring consistency in PIRADS scoring and biopsy methodologies.

Response:

Thank you for these valuable insights. In regard to the potential selection bias, we discussed it in detail in lines 250-260. We acknowledge that generalizability might be limited due to the validation being only internal. As noted in the conclusions, external validation is required before the nomogram can be applied in clinical practice.

In regard to MRI standardization and PIRADS consistency: As described in the methods (lines 71-73), no formal MRI data standardization was implemented. While this may be a limitation, as we state in the discussion, it reflects real-world clinical setting, where PIRADS scores are interpreted without centralized standardization. We have emphasized this point in the discussion section (lines 263-266). Although interobserver variability in MRI analysis is well recognized, our approach of including MRIs from various institutions provides a representative clinical sample. In the revised version of the manuscript, we put more emphasis on the issue of MRI interviewer variability (lines 261-263).

  1. The results are well presented, but the study does not analyze subgroups, such as biopsy-naïve vs. previously biopsied patients, which could impact outcomes. Include calibration plots to visually demonstrate model performance.

Thank you for this observation. Since biopsy-naïve status was not significantly associated with csPC diagnosis in univariable analysis (Table 2), we did not proceed with further subgroup stratification. Although biopsy history may influence individual csPC risk, in this specific subgroup (PSA < 10 ng/mL, DRE-negative), it did not. Calibration plots demonstrating model performance are included in Figure 4.

  1. The discussion overstates the clinical applicability of the nomogram without external validation. Certain findings, such as the role of lesion size, are dismissed without robust justification. 

Thank you for this important feedback. In the discussion and the conclusions, we emphasize, that external validation of the nomogram is warranted before clinical application of the model. Regarding lesion size, we did not dismiss its relevance but noted that it was not significantly associated with csPC in multivariable analysis and was therefore excluded from the final model. As detailed in the discussion, this might have been due to the predominance of small tumors in our cohort (Table 1).

  1. Figures lack sufficient annotations to explain key trends for readers unfamiliar with the statistical methods. Table 4 is dense and difficult to interpret without prior knowledge of thresholds.

Thank you for highlighting this. While we recognize potential difficulties that a reader unfamiliar with statistics may experience, we believe that the figures we present are clear to the readers familiar with ROC, calibration, or decision curves. The figures are commented on in the body of the text. Regarding Table 4, we appreciate your suggestion and have restructured it for improved readability. While we expanded the table horizontally, we understand that final formatting decisions may be made by the editor.

  1. The conclusion is overly optimistic regarding clinical implementation without discussing potential drawbacks, such as costs and accessibility of MRI-guided biopsies. Redundancy with the abstract’s concluding remarks.

Thank you for this comment. We did our best to avoid judgmental language in the conclusions section and we state that the limitations and need for external validation prevents from implementing the model into clinical practice at this time. In regard to the cost and accessibility of MRI-guided biopsies, this topic falls beyond the scope of this manuscript. As of current EAU guidelines, MRI is recommended before prostate biopsy. The biopsy procedure can be performed either with a software fusion system or can be a cognitive biopsy, depending on fusion hardware accessibility and urologist or patient preference. No superiority of any of these techniques has been shown in the literature. In our institution, all the biopsies were performed with a software fusion system, but the results may apply to cognitive biopsy patients as well.

Thank you for this comment. We aimed to avoid overstatements and emphasized the need for external validation before clinical application in the conclusions. In regard to the costs and accessibility of MRI-guided biopsies, this topic falls beyond the scope of our paper. Nevertheless, according to the current EAU guidelines, MRI is recommended before every biopsy (unless an advanced tumor is clinically apparent). Biopsies can be performed using either software fusion systems or cognitive techniques, with no proven superiority of one method over the other. In our institution, all biopsies were software-fusion guided, though the findings may apply to a cognitive biopsy setting as well.

Comments on the Quality of English Language

 There are occasional grammatical errors, such as inconsistent subject-verb agreement and misplaced modifiers (e.g., Calibration curves for both the model fitted on the whole training set and models fitted during the validation process demonstrated good calibration" could be simplified for better readability).

Certain sentences are unnecessarily complex, which detracts from the manuscript's clarity (e.g., "The association between 5-ARI use and increased risk of csPC is a notable finding" could be simplified to "The association between 5-ARI use and csPC risk is notable").

The manuscript frequently relies on passive voice, which can make statements less direct and impactful (e.g., Calibration curves were developed" could be rephrased as "We developed calibration curves").

We appreciate your feedback regarding language and clarity. We have revised the manuscript according to your suggestions.

Once again, we sincerely thank you for your thoughtful and constructive feedback. Your comments have significantly contributed to improving the clarity and overall quality of our manuscript. We hope that the revisions we have made address your concerns and meet the standards required for publication. We look forward to your further feedback and appreciate the time and effort you have invested in reviewing our work.

Round 2

Reviewer 1 Report

Comments and Suggestions for Authors

The authors addressed most of the concerns 

Reviewer 3 Report

Comments and Suggestions for Authors

All comments are justified and satisfactory